# A Universal Law of Robustness via Isoperimetry

**Sébastien Bubeck**
Microsoft Research
sebubeck@microsoft.com

**Mark Sellke**
Stanford University
msellke@stanford.edu

## Abstract

Classically, data interpolation with a parametrized model class is possible as long as the number of parameters is larger than the number of equations to be satisfied. A puzzling phenomenon in deep learning is that models are trained with many more parameters than what this classical theory would suggest. We propose a theoretical explanation for this phenomenon. We prove that for a broad class of data distributions and model classes, overparametrization is *necessary* if one wants to interpolate the data *smoothly*. Namely we show that *smooth* interpolation requires $d$ times more parameters than mere interpolation, where $d$ is the ambient data dimension. We prove this universal law of robustness for any smoothly parametrized function class with polynomial size weights, and any covariate distribution verifying isoperimetry (or a mixture thereof). In the case of two-layer neural networks and Gaussian covariates, this law was conjectured in prior work by Bubeck, Li and Nagaraj. We also give an interpretation of our result as an improved generalization bound for model classes consisting of smooth functions.

## 1 Introduction

Solving $n$ equations generically requires only $n$ unknowns[1]. However, the revolutionary deep learning methodology revolves around highly overparametrized models, with many more than $n$ parameters to learn from $n$ training data points. We propose an explanation for this enigmatic phenomenon, showing in great generality that finding a *smooth* function to fit $d$-dimensional data requires at least $nd$ parameters. In other words, overparametrization by a factor of $d$ is *necessary* for *smooth* interpolation, suggesting that perhaps the large size of the models used in deep learning is a *necessity* rather than a weakness of the framework. Another way to phrase the result is as a *tradeoff* between the size of a model (as measured by the number of parameters) and its "robustness" (as measured by its Lipschitz constant): either one has a small model (with $n$ parameters) which must then be non-robust, or one has a robust model (constant Lipschitz) but then it must be very large (with $nd$ parameters). Such a tradeoff was conjectured for the specific case of two-layer neural networks and Gaussian data in [BLN21]. Our result shows that in fact it is a *universal* phenomenon, which applies to essentially any parametrized function class (including in particular deep neural networks) as well as a much broader class of data distributions. As in [BLN21] we obtain an entire tradeoff curve between size and robustness: our universal law of robustness states that, for any function class smoothly parametrized by $p$ parameters, and for any $d$-dimensional dataset satisfying mild regularity conditions, any function in this class that fits the data *below the noise level* must have its (Euclidean) Lipschitz constant larger than $\sqrt{\frac{nd}{p}}$.

**Theorem 1** (Informal version of Theorem 3). *Let $\mathcal{F}$ be a class of functions from $\mathbb{R}^d \to \mathbb{R}$ and let $(x_i, y_i)_{i=1}^n$ be i.i.d. input-output pairs in $\mathbb{R}^d \times [-1, 1]$. Assume that:*

    *1. $\mathcal{F}$ admits a Lipschitz parametrization by $p$ real parameters, each of size at most $poly(n, d)$.*

---

[1]As in, for instance, the inverse function theorem in analysis or Bézout's theorem in algebraic geometry. See also [YSJ19, BELM20] for versions of this claim with neural networks.

35th Conference on Neural Information Processing Systems (NeurIPS 2021).

2. *The distribution $\mu$ of the covariates $x_i$ satisfies isoperimetry (or is a mixture theroeof).*

3. *The expected conditional variance of the output (i.e., the "noise level") is strictly positive, denoted $\sigma^2 := \mathbb{E}^\mu[Var[y|x]] > 0$.*

*Then, with high probability over the sampling of the data, one has simultaneously for all $f \in \mathcal{F}$:*

$$\frac{1}{n}\sum_{i=1}^n (f(x_i) - y_i)^2 \le \sigma^2 - \epsilon \ \Rightarrow \ \mathrm{Lip}(f) \ge \widetilde{\Omega}\left(\epsilon\sqrt{\frac{nd}{p}}\right).$$

**Remark 1.1.** For the distributions $\mu$ we have in mind, for instance uniform on the unit sphere, there exists with high probability some $O(1)$-Lipschitz function $f : \mathbb{R}^d \to \mathbb{R}$ satisfying $f(x_i) = y_i$ for all $i$. Indeed, with probability $1 - e^{-\Omega(d)}$ we have $||x_i - x_j|| \ge 1$ for all $1 \le i \ne j \le n$ so long as $n \le poly(d)$. In this case we may apply the Kirszbraun extension theorem to find a suitable $f$ regardless of the labels $y_i$. More explicitly we may fix a smooth bump function $g : \mathbb{R}^+ \to \mathbb{R}$ with $g(0) = 1$ and $g(x) = 0$ for $x \ge 1$, and then interpolate using the sum of radial basis functions

$$f(x) = \sum_{i=1}^n g(||x - x_i||)y_i.$$

In fact this construction requires only $p = n(d + 1)$ parameters to specify the values $(x_i, y_i)_{i \in [n]}$ and thus determine the function $f$. Hence $p = n(d + 1)$ parameters suffice for robust interpolation, i.e. Theorem 3 is essentially best possible for $L = O(1)$. A similar construction shows the same conclusion for any $p \in [\widetilde{\Omega}(n), nd]$, essentially tracing the entire tradeoff curve. This is because one can first project onto a fixed subspace of dimension $\tilde{d} = p/n$, and the projected inputs $x_i$ now have pairwise distances at least $\Omega\left(\sqrt{\tilde{d}/d}\right)$ with high probability. The analogous construction on the projected points now requires only $p = \tilde{d}n$ parameters and has Lipschitz constant $L = O\left(\sqrt{d/\tilde{d}}\right) = O\left(\sqrt{\frac{nd}{p}}\right)$.

## 1.1 Speculative implication for real data

To put Theorem 1 in context, we compare to the empirical results presented in [MMS$^+$18]. In the latter work, they consider the MNIST dataset which consists of $n = 6 \times 10^4$ images in dimension $28^2 = 784$. They trained robustly different architectures, and reported in Figure 4 the size of the architecture versus the obtained robust test accuracy (third plot from the left). One can see a sharp transition from roughly 10% accuracy to roughly 90% accuracy at around $2 \times 10^5$ parameters (capacity scale 4 in their notation). Moreover the robust accuracy keeps climbing up with more parameters, to roughly 95% accuracy at roughly $3 \times 10^6$ parameters.

How can we compare these numbers to the law of robustness? There are a number of difficulties that we discuss below, and we emphasize that this discussion is highly speculative in nature, though we find that, with a few leaps of faith, our universal law of robustness sheds light on the potential parameter regimes of interest for robust deep learning.

The first difficulty is to evaluate the "correct" dimension of the problem. Certainly the number of pixels per image gives an upper bound, however one expects that the data lies on something like a lower dimensional sub-manifold. Optimistically, we hope that Theorem 1 will continue to apply for an appropriate *effective dimension* which may be rather smaller than the literal number of pixels. This hope is partially justified by the fact that isoperimetry holds in many less-than-picturesque situations, some of which are stated in the next subsection.

The next difficulty is to estimate/interpret the noise value $\sigma^2$. From a theoretical point of view, this noise assumption is necessary for otherwise there could exist a smooth classifier with perfect accuracy in $\mathcal{F}$, defeating the point of any lower bound on the size of $\mathcal{F}$. We tentatively would like to think of $\sigma^2$ as capturing the contribution of the "difficult" part of the learning problem, that is $\sigma^2$ could be thought of as the non-robust generalization error of reasonably good models, so a couple

of % of error in the case of MNIST. With that interpretation, one gets "below the noise level" in MNIST with a training error of a couple of %. We believe that versions of the law of robustness might hold without noise; these would need to go beyond representational power and consider the dynamics of learning algorithms.

Finally another subtlety to interpret the empirical results of [MMS$^+$18] is that there is a mismatch between what they measure and our quantities of interest. Namely the law of robustness talks about two things: the training error, and the worst-case robustness (i.e., the Lipschitz constant). On the other hand [MMS$^+$18] measures the *robust generalization error*. Understanding the interplay between those three quantities is a fantastic open problem. Here we take the perspective that a small robust generalization error should imply a small training error and a small Lipschitz constant. Another important mismatch is that we stated our universal law of robustness for Lipschitzness in $\ell_2$, while the experiments in [MMS$^+$18] are for robustness in $\ell_\infty$. We believe that a variant of the law of robustness remains true for $\ell_\infty$, a belief again partially justified by how broad isoperimetry is (see next subsection).

With all the caveats described above, we can now look at the numbers as follows: in the [MMS$^+$18] experiments, smooth models with accuracy below the noise level are attained with a number of parameters somewhere in the range $2 \times 10^5 - 3 \times 10^6$ parameters (possibly even larger depending on the interpretation of the noise level), while the law of robustness would predict any such model must have at least $nd$ parameters, and this latter quantity should be somewhere in the range $10^6 - 10^7$ (corresponding to an effective dimension between 15 and 150). While far from perfect, the law of robustness prediction is far more accurate than the classical rule of thumb # parameters $\simeq$ # equations (which here would predict a number of parameters of the order $10^4$).

Perhaps more interestingly, one could apply a similar reasoning to the ImageNet dataset, which consists of $1.4 \times 10^7$ images of size roughly $2 \times 10^5$. Estimating that the effective dimension is a couple of order of magnitudes smaller than this size, the law of robustness predicts that to obtain good robust models on ImageNet one would need at least $10^{10} - 10^{11}$ parameters. This number is larger than the size of current neural networks trained robustly for this task, which sports between $10^8 - 10^9$ parameters. Thus, we arrive at the tantalizing possibility that robust models for ImageNet do not exist yet simply because we are a couple orders of magnitude off in the current scale of neural networks trained for this task.

## 1.2   Related work

Theorem 1 is a direct follow-up to the conjectured law of robustness in [BLN21] for (arbitrarily weighted) two-layer neural networks with Gaussian data. Our result does not actually prove their conjecture, because we assume here polynomially bounded weights. While this assumption is reasonable from a practical perspective, it remains mathematically interesting to prove the full conjecture for the two-layer case. We prove however in Section A that the polynomial weights assumption is necessary as soon as one considers three-layer neural networks. Let us also mention the [GCL$^+$19, Theorem 6.1] which showed a lower bound $\Omega(nd)$ on the VC dimension of any function class which can robustly interpolate *arbitrary* labels on *all* well-separated input sets $(x_1, \ldots, x_n)$. We also note that a relation between high-dimensional phenomenon such as concentration and adversarial examples has been hypothesized before, such as in [GMF$^+$18].

In addition to [MMS$^+$18], several recent works have experimentally studied the relationship between a neural network scale and its achieved robustness, see e.g., [NBA$^+$18, XY20, GQU$^+$20]. It has been consistently reported that larger networks help tremendously for robustness, beyond what is typically seen for classical non-robust accuracy. We view our universal law of robustness as putting this empirical observation on a more solid footing: scale is actually *necessary* to achieve robustness.

The law of robustness setting is closely related to the interpolation setting: in the former case one considers models optimizing "beyond the noise level", while in the latter case one studies models with perfect fit on the training data. The study of generalization in this in-

terpolation regime has been a central focus of learning theory in the last few years (see e.g., [BHMM19, MM19, BLLT20, NKB$^+$20]), as it seemingly contradicts classical theory about regularization. More broadly though, generalization remains a mysterious phenomon in deep learning, and the exact interplay between the law of robustness' setting (interpolation regime/worst-case robustness) and (robust) generalization error is a fantastic open problem. Interestingly, we note that one could potentially avoid the conclusion of the law of robustness (that is, that large models are necessary for robustness), with early stopping methods that could stop the optimization once the noise level is reached. In fact, this theoretically motivated suggestion has already been empirically tested and confirmed in the recent work [RWK20], showing again a close tie between the conclusions one can draw from the law of robustness and actual practical settings.

Classical lower bounds on the gradient of a function include Poincaré type inequalities, but they are of a qualitately different nature compared to the law of robustness lower bound. We recall that a measure $\mu$ on $\mathbb{R}^d$ satisfies a Poincaré inequality if for any function $f$, one has $\mathbb{E}^\mu[\|\nabla f\|^2] \geq C \cdot \mathrm{Var}(f)$ (for some constant $C > 0$). In our context, such a lower bound for an interpolating function $f$ has essentially no consequence since the variance $f$ could be exponentially small. In fact this is tight, as one easily use similar constructions to those in [BLN21] to show that one can interpolate with an exponentially small expected norm squared of the gradient (in particular it is crucial in the law of robustness to consider the Lipschitz constant, i.e., the supremum of the norm of the gradient). On the other hand, our isoperimetry assumption is related to a certain strenghtening of the Poincaré inequality known as log-Sobolev inequality (see e.g., [Led01]). If the covariate measure satisfies only a Poincaré inequality, then we could prove a weaker law of robustness of the form $\mathrm{Lip} \gtrsim \frac{n\sqrt{d}}{p}$ (using for example the concentration result obtained in [BL97]). For the case of two-layer neural networks there is another natural notion of smoothness (different from $\ell_p$ norms of the gradient) that can be considered, known as the Barron norm. In [BELM20] it is shown that for such a notion of smoothness there is no tradeoff à la the law of robustness, namely one can simultaneously be optimal both in terms of Barron norm and in terms of the network size. More generally, it is an interesting challenge to understand for which notions of smoothness there is a tradeoff with size.

## 1.3 Isoperimetry

Concentration of measure and isoperimetry are perhaps the most ubiquitous features of high-dimensional geometry. In short, they assert in many cases that Lipschitz functions on high-dimensional space concentrate tightly around their mean. Our result assumes that the distribution $\mu$ of the covariates $x_i$ satisfies such an inequality in the following sense.

**Definition 1.1.** *A probability measure $\mu$ on $\mathbb{R}^d$ satisfies c-isoperimetry if for any bounded L-Lipschitz $f : \mathbb{R}^d \to \mathbb{R}$, and any $t \geq 0$,*

$$\mathbb{P}[|f(x) - \mathbb{E}[f]| \geq t] \leq 2e^{-\frac{dt^2}{2cL^2}}. \tag{1.1}$$

In general, if a scalar random variable $X$ satisfies $\mathbb{P}[|X| \geq t] \leq 2e^{-t^2/C}$ then we say $X$ is $C$-subgaussian. Hence isoperimetry states that the output of any Lipschitz function is $O(1)$-subgaussian under suitable rescaling. Distributions satisfying $O(1)$-isoperimetry include high dimensional Gaussians $\mu = \mathcal{N}\left(0, \frac{I_d}{d}\right)$ and uniform distributions on spheres and hypercubes (normalized to have diameter 1). Isoperimetry also holds for mild perturbations of these idealized scenarios, including[2]:

- The sum of a Gaussian and an independent random vector of small norm [CCNW21].
- Strongly log-concave measures in any normed space [BL00, Proposition 3.1].
- Manifolds with positive Ricci curvature [Gro86, Theorem 2.2].

Due to the last condition above, we believe our results are realistic even under the *manifold hypothesis* that high-dimensional data tends to lie on a lower-dimensional submanifold. This viewpoint on learning has been studied for decades, see e.g. [HS89, KL93, RS00, TDSL00, NM10, FMN16].

---

[2]The first two examples satisfy a logarithmic Sobolev inequality, which implies isoperimetry [Led99, Proposition 2.3].

We also note that our formal theorem (Theorem 3) actually applies to distributions that can be written as a mixture of distributions satisfying isoperimetry. Let us also point out that from a technical perspective, our proof is not tied to the Euclidean norm and applies essentially whenever Definition 1.1 holds. The main difficulty in extending the law of robustness to e.g. the earth-mover distance seems to be identifying realistic cases which satisfy isoperimetry.

Our proofs will repeatedly use the following simple fact:

**Proposition 1.2.** *[Ver18, Proposition 2.6.1],[vH14, Exercise 3.1] If $X_1, \ldots, X_n$ are independent, $C$-subgaussian, with mean $0$, then $\frac{1}{\sqrt{n}} \sum_{i=1}^{n} X_i$ is $18C$-subgaussian.*

## 2 A finite approach to the law of robustness

For the function class of two-layer neural networks, [BLN21] investigated several approaches to prove the law of robustness. At a high level, the proof strategies there relied on various ways to measure how "large" the set of two-layer neural networks can be (specifically, they tried a geometric approach based on relating to multi-index models, a statistical approach based on the Rademacher complexity, and an algebraic approach for the case of polynomial activations).

In this work we take here a different route: we shift the focus from the function class $\mathcal{F}$ to an *individual* function $f \in \mathcal{F}$. Namely, our proof starts by asking the following question: for a fixed function $f$, what is the probability that it would give a good approximate fit on the (random) data? For simplicity, consider for a moment the case where we require $f$ to actually interpolate the data (i.e., perfect fit), and say that $y_i$ are random $\pm 1$ labels. The key insight is that isoperimetry implies that *either* the $0$-level set of $f$ *or* the $1$-level set of $f$ must have probability smaller than $\exp\left(-\frac{d}{\mathrm{Lip}(f)^2}\right)$. Thus, the probability that $f$ fits all the $n$ points is at most $\exp\left(-\frac{nd}{\mathrm{Lip}(f)^2}\right)$ so long as both labels $y_i \in \{-1, 1\}$ actually appear a constant fraction of the time. In particular, using an union bound[3], for a finite function class $\mathcal{F}$ of size $N$ with $L$-Lipschitz functions, the probability that there exists a function $f \in \mathcal{F}$ fitting the data is at most

$$N \exp\left(-\frac{nd}{L^2}\right) = \exp\left(\log(N) - \frac{nd}{L^2}\right).$$

Thus we see that, if $L \ll \sqrt{\frac{nd}{\log(N)}}$, then the probability of finding a fitting function in $\mathcal{F}$ is very small. This basically concludes the proof, since via a standard discretization argument, for a smoothly parametrized family with $p$ (bounded) parameters one expects $\log(N) = \tilde{O}(p)$.

We now give the formal proof, which applies in particular to approximate fit rather than exact fit in the argument above. The only difference is that we will identify a well-chosen subgaussian random variable in the problem. We start with the finite function class case:

**Theorem 2.** *Let $(x_i, y_i)$ be i.i.d. input-output pairs in $\mathbb{R}^d \times [-1, 1]$ such that:*

1. *The distribution $\mu$ of the covariates $x_i$ can be written as $\mu = \sum_{\ell=1}^{k} \alpha_\ell \mu_\ell$, where each $\mu_\ell$ satisfies $c$-isoperimetry and $\alpha_\ell \geq 0, \sum_{\ell=1}^{k} \alpha_\ell = 1$.*

2. *The expected conditional variance of the output is strictly positive, denoted $\sigma^2 := \mathbb{E}^\mu[Var[y|x]] > 0$.*

*Then one has:*

$$\mathbb{P}\left(\exists f \in \mathcal{F} : \frac{1}{n} \sum_{i=1}^{n} (y_i - f(x_i))^2 \leq \sigma^2 - \epsilon\right)$$

$$\leq 4k \exp\left(-\frac{n\epsilon^2}{8^3 k}\right) + 2 \exp\left(\log(|\mathcal{F}|) - \frac{\epsilon^2 nd}{9^4 cL^2}\right).$$

---

[3]In this informal argument we ignore the possibility that the labels $y_i$ are not well-balanced. Note that the probability of this rare event is not amplified by a union bound over $f \in \mathcal{F}$.

We start with a lemma showing that, to optimize beyond the noise level one must necessarily correlate with the noise part of the labels. In what follows we denote $g(x) = \mathbb{E}[y|x]$ for the target function, and $z_i = y_i - g(x_i)$ for the noise part of the observed labels (namely $y_i$ is the sum of the target function $g(x_i)$ and the noise term $z_i$).

**Lemma 2.1.** *One has*

$$\mathbb{P}\left(\exists f \in \mathcal{F}: \frac{1}{n}\sum_{i=1}^{n}(y_i - f(x_i))^2 \leq \sigma^2 - \epsilon\right) \leq 2\exp\left(-\frac{n\epsilon^2}{8^3}\right) + \mathbb{P}\left(\exists f \in \mathcal{F}: \frac{1}{n}\sum_{i=1}^{n}f(x_i)z_i \geq \frac{\epsilon}{4}\right).$$

*Proof.* The sequence $(z_i^2)$ is i.i.d., with mean $\sigma^2$, and such that $|z_i|^2 \leq 4$. Thus Hoeffding's inequality yields:

$$\mathbb{P}\left(\frac{1}{n}\sum_{i=1}^{n}z_i^2 \leq \sigma^2 - \frac{\epsilon}{6}\right) \leq \exp\left(-\frac{n\epsilon^2}{8^3}\right). \tag{2.1}$$

On the other hand the sequence $(z_i g(x_i))$ is i.i.d., with mean 0 (since $\mathbb{E}[z_i|x_i] = 0$), and such that $|z_i g(x_i)| \leq 2$. Thus Hoeffding's inequality yields:

$$\mathbb{P}\left(\frac{1}{n}\sum_{i=1}^{n}z_i g(x_i) \leq -\frac{\epsilon}{6}\right) \leq \exp\left(-\frac{n\epsilon^2}{8^3}\right). \tag{2.2}$$

Let us write $Z = \frac{1}{\sqrt{n}}(z_1,\ldots,z_n), G = \frac{1}{\sqrt{n}}(g(x_1),\ldots,g(x_n))$, and $F = \frac{1}{\sqrt{n}}(f(x_1),\ldots,f(x_n))$. We claim that if $\|Z\|^2 \geq \sigma^2 - \frac{\epsilon}{6}$ and $\langle Z, G\rangle \geq -\frac{\epsilon}{6}$, then for any $f \in \mathcal{F}$ one has

$$\|G + Z - F\|^2 \leq \sigma^2 - \epsilon \Rightarrow \langle F, Z\rangle \geq \frac{\epsilon}{4}.$$

This claim together with (2.1) and (2.2) conclude the proof. On the other hand the claim itself directly follows from:

$$\sigma^2 - \epsilon \geq \|G + Z - F\|^2 = \|Z + G - F\|^2 = \|Z\|^2 + 2\langle Z, G - F\rangle + \|G - F\|^2 \geq \sigma^2 - \frac{\epsilon}{2} - 2\langle Z, F\rangle.$$

$\square$

We can now proceed to the proof of Theorem 2:

*Proof.* First note that without loss of generality we can assume that the range of any function in $\mathcal{F}$ is included in $[-1, 1]$ (indeed clipping the values improves both the fit to any $y \in [-1, 1]$ and the Lipschitz constant). We also assume wlog that all functions in $\mathcal{F}$ are $L$-Lipschitz.

For clarity let us start with the case $k = 1$. By the isoperimetry assumption we have that $\sqrt{\frac{d}{c}}\frac{f(x_i) - \mathbb{E}[f]}{L}$ is 1-subgaussian. Since $|z_i| \leq 2$, we also have that $\sqrt{\frac{d}{c}}\frac{(f(x_i) - \mathbb{E}[f])z_i}{L}$ is 4-subgaussian. Moreover, the latter random variable has zero-mean since $\mathbb{E}[z|x] = 0$. Thus by Proposition 1.2 we have:

$$\mathbb{P}\left(\sqrt{\frac{d}{cnL^2}}\sum_{i=1}^{n}(f(x_i) - \mathbb{E}[f])z_i \geq t\right) \leq 2\exp\left(-(t/9)^2\right).$$

We rewrite the above as:

$$\mathbb{P}\left(\frac{1}{n}\sum_{i=1}^{n}(f(x_i) - \mathbb{E}[f])z_i \geq \frac{\epsilon}{8}\right) \leq 2\exp\left(-\frac{\epsilon^2 nd}{9^4 cL^2}\right). \tag{2.3}$$

Since we assumed that the range of the functions is in $[-1, 1]$ we have $\mathbb{E}[f] \in [-1, 1]$ and hence:

$$\mathbb{P}\left(\exists f \in \mathcal{F}: \frac{1}{n}\sum_{i=1}^{n}\mathbb{E}[f]z_i \geq \frac{\epsilon}{8}\right) \leq \mathbb{P}\left(\left|\frac{1}{n}\sum_{i=1}^{n}z_i\right| \geq \frac{\epsilon}{8}\right). \tag{2.4}$$

(This step is the analog of requiring the labels $y_i$ to be well-balanced in the example of perfect interpolation.) By Hoeffding's inequality, the above quantity is smaller than $2\exp(-n\epsilon^2/8^3)$ (recall that $|z_i| \leq 2$). Thus we obtain with an union bound:

$$\mathbb{P}\left(\exists f \in \mathcal{F} : \frac{1}{n}\sum_{i=1}^n f(x_i)z_i \geq \frac{\epsilon}{4}\right) \leq |\mathcal{F}| \cdot \mathbb{P}\left(\frac{1}{n}\sum_{i=1}^n (f(x_i) - \mathbb{E}[f])z_i \geq \frac{\epsilon}{8}\right) + \mathbb{P}\left(\left|\frac{1}{n}\sum_{i=1}^n z_i\right| \geq \frac{\epsilon}{8}\right)$$

$$\leq 2|\mathcal{F}| \cdot \exp\left(-\frac{\epsilon^2 nd}{9^4 cL^2}\right) + 2\exp\left(-\frac{n\epsilon^2}{8^3}\right).$$

Together with Lemma 2.1 this concludes the proof for $k = 1$.

We now turn to the case $k > 1$. We first sample the mixture component $\ell_i \in [k]$ for each data point $i \in [n]$, and we now reason conditioned on these mixture components. Let $S_\ell \subset [n]$ be the set of data points sampled from mixture component $\ell \in [k]$, that is $x_i, i \in S_\ell$, is i.i.d. from $\mu_\ell$. We now have that $\sqrt{\frac{d}{c}}\frac{f(x_i) - \mathbb{E}^{\mu_{\ell_i}}[f]}{L}$ is 1-subgaussian (notice that the only difference is that now we need to center by $\mathbb{E}^{\mu_{\ell_i}}[f]$, which depends on the mixture component). In particular using the same reasoning as for (2.4) we obtain (crucially note that Proposition 1.2 does not require the random variables to be identically distributed):

$$\mathbb{P}\left(\frac{1}{n}\sum_{i=1}^n (f(x_i) - \mathbb{E}^{\mu_{\ell_i}}[f])z_i \geq \frac{\epsilon}{8}\right) \leq 2\exp\left(-\frac{\epsilon^2 nd}{9^4 cL^2}\right). \tag{2.5}$$

Next we want to appropriately modify (2.4). To do so note that:

$$\max_{m_1,\ldots,m_k \in [-1,1]} \sum_{i=1}^n m_{\ell_i} z_i = \sum_{\ell=1}^k \left|\sum_{i \in S_\ell} z_i\right|,$$

so that we can rewrite (2.4) as:

$$\mathbb{P}\left(\exists f \in \mathcal{F} : \frac{1}{n}\sum_{i=1}^n \mathbb{E}^{\mu_{\ell_i}}[f]z_i \geq \frac{\epsilon}{8}\right) \leq \mathbb{P}\left(\frac{1}{n}\sum_{\ell=1}^k \left|\sum_{i \in S_\ell} z_i\right| \geq \frac{\epsilon}{8}\right).$$

Now note that $\sum_{\ell=1}^k \sqrt{|S_\ell|} \leq \sqrt{nk}$ and thus we have:

$$\mathbb{P}\left(\frac{1}{n}\sum_{\ell=1}^k \left|\sum_{i \in S_\ell} z_i\right| \geq \frac{\epsilon}{8}\right) \leq \mathbb{P}\left(\sum_{\ell=1}^k \left|\sum_{i \in S_\ell} z_i\right| \geq \frac{\epsilon}{8}\sqrt{\frac{n}{k}}\sum_{\ell=1}^k \sqrt{|S_\ell|}\right) \leq \sum_{\ell=1}^k \mathbb{P}\left(\left|\sum_{i \in S_\ell} z_i\right| \geq \frac{\epsilon}{8}\sqrt{\frac{n}{k}}\sqrt{|S_\ell|}\right).$$

Finally by Hoeffding's inequality, we have for any $\ell \in [k]$, $\mathbb{P}\left(\left|\sum_{i \in S_\ell} z_i\right| \geq t\sqrt{|S_\ell|}\right) \leq 2\exp\left(-\frac{t^2}{8}\right)$, and thus the last display is bounded from above by $2k\exp\left(-\frac{n\epsilon^2}{8^3 k}\right)$. The proof can now be concluded as in the case $k = 1$. $\square$

Finally we can now state and prove the formal version of the informal Theorem 1 from the introduction.

**Theorem 3.** *Let $\mathcal{F}$ be a class of functions from $\mathbb{R}^d \to \mathbb{R}$ and let $(x_i, y_i)_{i=1}^n$ be i.i.d. input-output pairs in $\mathbb{R}^d \times [-1, 1]$. Fix $\epsilon, \delta \in (0, 1)$. Assume that:*

1. *The function class can be written as $\mathcal{F} = \{f_{\boldsymbol{w}}, \boldsymbol{w} \in \mathcal{W}\}$ with $\mathcal{W} \subset \mathbb{R}^p$, $\text{diam}(\mathcal{W}) \leq W$ and for any $\boldsymbol{w}_1, \boldsymbol{w}_2 \in \mathcal{W}$,*

$$||f_{\boldsymbol{w}_1} - f_{\boldsymbol{w}_2}||_\infty \leq J||\boldsymbol{w}_1 - \boldsymbol{w}_2||.$$

2. *The distribution $\mu$ of the covariates $x_i$ can be written as $\mu = \sum_{\ell=1}^k \alpha_\ell \mu_\ell$, where each $\mu_\ell$ satisfies $c$-isoperimetry, $\alpha_\ell \geq 0$, $\sum_{\ell=1}^k \alpha_\ell = 1$, and $k$ is such that $9^4 k \log(8k/\delta) \leq n\epsilon^2$.*

3. *The expected conditional variance of the output is strictly positive, denoted $\sigma^2 := \mathbb{E}^\mu[Var[y|x]] > 0$.*

*Then, with probability at least $1 - \delta$ with respect to the sampling of the data, one has simultaneously for all $f \in \mathcal{F}$:*

$$\frac{1}{n}\sum_{i=1}^{n}(f(x_i) - y_i)^2 \leq \sigma^2 - \epsilon \implies \mathrm{Lip}(f) \geq \frac{\epsilon}{2^9\sqrt{c}}\sqrt{\frac{nd}{p\log(60WJ\epsilon^{-1}) + \log(4/\delta)}}\,.$$

*Proof.* Define $\mathcal{W}_L \subseteq \mathcal{W}$ by $\mathcal{W}_L = \{\boldsymbol{w} \in \mathcal{W} : \mathrm{Lip}(f_{\boldsymbol{w}}) \leq L\}$. Denote $\mathcal{W}_{L,\epsilon}$ for an $\frac{\epsilon}{6J}$-net of $\mathcal{W}_L$. We have in particular $|\mathcal{W}_\epsilon| \leq (60WJ\epsilon^{-1})^p$. We apply Theorem 2 to $\mathcal{F}_{L,\epsilon} = \{f_{\boldsymbol{w}}, \boldsymbol{w} \in \mathcal{W}_{L,\epsilon}\}$:

$$\mathbb{P}\left(\exists f \in \mathcal{F}_{L,\epsilon} : \frac{1}{n}\sum_{i=1}^{n}(y_i - f(x_i))^2 \leq \sigma^2 - \frac{\epsilon}{2} \text{ and } \mathrm{Lip}(f) \leq 2L\right)$$

$$\leq 4k\exp\left(-\frac{n\epsilon^2}{9^4k}\right) + 2\exp\left(p\log(60WJ\epsilon^{-1}) - \frac{\epsilon^2nd}{8^6cL^2}\right)\,.$$

Observe that if $\|f - g\|_\infty \leq \frac{\epsilon}{6}$ and $\|y\|_\infty, \|f\|_\infty, \|g\|_\infty \leq 1$, then $\frac{1}{n}\sum_{i=1}^{n}(y_i - f(x_i))^2 \leq \frac{\epsilon}{2} + \frac{1}{n}\sum_{i=1}^{n}(y_i - g(x_i))^2$. (We may again assume without loss of generality that all functions in $\mathcal{F}$ map to $[-1,1]$.) Thus we obtain for any $L > 0$:

$$\mathbb{P}\left(\exists f \in \mathcal{F} : \frac{1}{n}\sum_{i=1}^{n}(y_i - f(x_i))^2 \leq \sigma^2 - \epsilon \text{ and } \mathrm{Lip}(f) \leq L\right)$$

$$\leq 4k\exp\left(-\frac{n\epsilon^2}{9^4k}\right) + 2\exp\left(p\log(60WJ\epsilon^{-1}) - \frac{\epsilon^2nd}{8^6cL^2}\right)\,.$$

The first assumption ensures that for any $\boldsymbol{w} \in \mathcal{W}_L$, there is $\boldsymbol{w}' \in \mathcal{W}_{L,\epsilon}$ with $\|f_{\boldsymbol{w}} - f_{\boldsymbol{w}'}\|_\infty \leq \frac{\epsilon}{6}$. The second assumption shows the probability just above is at most $\delta$ when $L = \frac{\epsilon}{2^9\sqrt{c}}\sqrt{\frac{nd}{p\log(60WJ\epsilon^{-1})+\log(4/\delta)}}$. This concludes the proof. $\qquad\square$

# 3 Deep neural networks

We now specialize the law of robustness (Theorem 3) to multi-layer neural networks. We consider a rather general class of depth $D$ neural networks described as follows. First, we require that the neurons are partitioned into layers $\mathcal{L}_1, \ldots, \mathcal{L}_D$, and that all connections are from $\mathcal{L}_i \to \mathcal{L}_j$ for some $i < j$. This includes the basic feed-forward case in which only connections $\mathcal{L}_i \to \mathcal{L}_{i+1}$ are used as well as more general skip connections. We specify (in the natural way) a neural network by matrices $W_j$ of shape $|\mathcal{L}_j| \times \sum_{i<j}|\mathcal{L}_i|$ for each $1 \leq j \leq D$, as well as 1-Lipschitz non-linearities $\sigma_{j,\ell}$ and scalar biases $b_{j,\ell}$ for each $(j,\ell)$ satisfying $\ell \in |\mathcal{L}_j|$. We use fixed non-linearities $\sigma_{j,\ell}$ as well as a fixed architecture, in the sense that each matrix entry $W_j[k,\ell]$ is either always 0 or else it is variable (and similarly for the bias terms).

To match the notation of Theorem 3, we identify the parametrization in terms of the matrices $(W_j)$ and bias terms $(b_{j,\ell})$ to a single $p$-dimensional vector $\boldsymbol{w}$ as follows. A variable matrix entry $W_j[k,\ell]$ is set to $w_{a(j,k,\ell)}$ for some fixed index $a(j,k,\ell) \in [p]$, and a variable bias term $b_{j,\ell}$ is set to $w_{a(j,\ell)}$ for some $a(j,\ell) \in [p]$. Thus we now have a parametrization $\boldsymbol{w} \in \mathbb{R}^p \mapsto f_{\boldsymbol{w}}$ where $f_{\boldsymbol{w}}$ is the neural network represented by the parameter vector $\boldsymbol{w}$. Importantly, note that our formulation allows for weight sharing (in the sense that a shared weight is counted only as a single parameter). For example, this is important to obtain an accurate count of the number of parameters in convolutional architectures.

In order to apply Theorem 3 to this class of functions we need to estimate the Lipschitz constant of the parametrization $\boldsymbol{w} \mapsto f_{\boldsymbol{w}}$. To do this we introduce three more quantities. First, we shall assume that all the parameters are bounded in magnitude by $W$, that is we consider the set of neural networks parametrized by $\boldsymbol{w} \in [-W,W]^p$. Next, for the architecture under consideration, denote $Q$ for the maximum number of matrix entries/bias terms that are tied to a single parameter $w_a$ for some $a \in [p]$. Finally we define

$$B(\boldsymbol{w}) = \prod_{j\in[D]}\max(\|W_j\|_{op}, 1).$$

Observe that $B(\boldsymbol{w})$ is an upper bound on the Lipschitz constant of the network itself, i.e., the map $x \mapsto f_{\boldsymbol{w}}(x)$. It turns out that a uniform control on it also controls the Lipschitz constant of the *parametrization* $\boldsymbol{w} \mapsto f_{\boldsymbol{w}}$. Namely we have the following lemma:

**Lemma 3.1.** *Let $x \in \mathbb{R}^d$ such that $\|x\| \leq R$, and $\boldsymbol{w}_1, \boldsymbol{w}_2 \in \mathbb{R}^p$ such that $B(\boldsymbol{w}_1), B(\boldsymbol{w}_2) \leq \overline{B}$. Then one has*

$$|f_{\boldsymbol{w}_1}(x) - f_{\boldsymbol{w}_2}(x)| \leq \overline{B}^2 QR\sqrt{p}\|\boldsymbol{w}_1 - \boldsymbol{w}_2\|.$$

*Moreover for any $\boldsymbol{w} \in [-W, W]^p$ with $W \geq 1$, one has*

$$B(\boldsymbol{w}) \leq (W\sqrt{pQ})^D.$$

*Proof.* Fix an input $x$ and define $g_x$ by $g_x(\boldsymbol{w}) = f_{\boldsymbol{w}}(x)$. A standard gradient calculation for multi-layer neural networks directly shows that $\|\nabla g_x(\boldsymbol{w})\|_\infty \leq B(\boldsymbol{w})QR$ so that $\|\nabla g_x(\boldsymbol{w})\| \leq B(\boldsymbol{w})QR\sqrt{p}$. Since the matrix operator norm is convex (and nonnegative) it follows that $B(\boldsymbol{w}) \leq B(\boldsymbol{w}_1)B(\boldsymbol{w}_2) \leq \overline{B}^2$ on the entire segment $[\boldsymbol{w}_1, \boldsymbol{w}_2]$ by multiplying over layers. Thus $\|\nabla g_x(\boldsymbol{w})\| \leq \overline{B}^2 QR\sqrt{p}$ on that segment, which concludes the proof of the first claimed inequality. The second claimed inequality follows directly from $\|W_j\|_{op} \leq \|W_j\|_2 \leq W\sqrt{pQ}$. $\qquad\square$

Lemma 3.1 shows that when applying Theorem 3 to our class of neural networks one can always take $J = R(WQp)^D$ (assuming that the covariate measure $\mu$ is supported on the ball of radius $R$). Thus in this case the law of robustness (under the assumptions of Theorem 3) directly states that with high probability, any neural network in our class that fits the training data well below the noise level must also have:

$$\mathrm{Lip}(f) \geq \tilde{\Omega}\left(\sqrt{\frac{nd}{Dp}}\right), \tag{3.1}$$

where $\tilde{\Omega}$ hides logarithmic factors in $W, p, R, Q$, and the probability of error $\delta$. Thus we see that the law of robustness, namely that the number of parameters should be at least $nd$ for a smooth model with low training error, remains intact for constant depth neural networks. If taken at face value, the lower bound (3.1) suggests that it is better in practice to distribute the parameters towards *depth* rather than *width*, since the lower bound is decreasing with $D$. On the other hand, we note that (3.1) can be strengthened to:

$$\mathrm{Lip}(f) \geq \tilde{\Omega}\left(\sqrt{\frac{nd}{p\log(\overline{B})}}\right), \tag{3.2}$$

for the class of neural networks such that $B(\boldsymbol{w}) \leq \overline{B}$. In other words the dependency on the depth all but disappears by simply assuming that the quantity $B(\boldsymbol{w})$ (a natural upper bound on the Lipschitz constant of the network) is polynomially controlled. Interestingly many works have suggested to keep $B(\boldsymbol{w})$ under control, either for regularization purpose (for example [BFT17] relates $B(\boldsymbol{w})$ to the Rademacher complexity of multi-layer neural networks) or to simply control gradient explosion during training, see e.g., [ASB16, CBG+17, MHRB17, MKKY18, JCC+19, YM17]. Moreover, in addition to being well-motivated in practice, the assumption that $\overline{B}$ is polynomially controlled seems also somewhat unavoidable in theory, since $B(\boldsymbol{w})$ is an *upper bound* on the Lipschitz constant $\mathrm{Lip}(f_{\boldsymbol{w}})$. Thus a theoretical construction showing that the lower bound in (3.1) is tight (at some large depth $D$) would necessarily need to have an exponential gap between $\mathrm{Lip}(f_{\boldsymbol{w}})$ and $B(\boldsymbol{w})$. We are not aware of any such example, and it would be interesting to fully elucidate the role of depth in the law of robustness (particularly if it could give recommendation on how to best distribute parameters in a neural network).

## 4 Generalization Perspective

The law of robustness can be phrased in a slightly stronger way, as a generalization bound for classes of Lipschitz functions based on data-dependent Rademacher complexity. In particular, this perspective applies to any Lipschitz loss function, whereas our analysis in the main text was specific to the squared loss. We define the data-dependent Rademacher complexity $\mathrm{Rad}_{n,\mu}(\mathcal{F})$ by

$$\mathrm{Rad}_{n,\mu}(\mathcal{F}) = \frac{1}{n}\mathbb{E}^{\sigma_i, x_i}\left[\sup_{f \in \mathcal{F}}\left|\sum_{i=1}^{n}\sigma_i f(x_i)\right|\right] \tag{4.1}$$

where the values $(\sigma_i)_{i \in [n]}$ are i.i.d. symmetric Rademacher variables in $\{-1, 1\}$ while the values $(x_i)_{i \in [n]}$ are i.i.d. samples from $\mu$.

**Lemma 4.1.** *Suppose $\mu = \sum_{i=1}^{k} \alpha_i \mu_i$ is a mixture of c-isoperimetric distributions. For finite $\mathcal{F}$ consisting of L-Lipschitz $f$ with $|f(x)| \leq 1$ for all $(f, x) \in \mathcal{F} \times \mathbb{R}^d$, we have*

$$Rad_{n,\mu}(\mathcal{F}) \leq O\left(\max\left(\sqrt{\frac{k}{n}}, L\sqrt{\frac{c\log(|\mathcal{F}|)}{nd}}\right)\right). \tag{4.2}$$

The proof is identical to that of Theorem 2. Note that $Rad_{n,\mu}(\mathcal{F})$ simply measures the ability of functions in $\mathcal{F}$ to correlate with random noise. Using standard machinery it implies the following generalization bound:

**Corollary 4.2.** *For any loss function $\ell(t, y)$ which is bounded and 1-Lipschitz in its first argument and any $\delta \in [0, 1]$, in the setting of Lemma 4.1 we have with probability at least $1 - \delta$ the uniform convergence bound:*

$$\sup_{f \in \mathcal{F}} \left| \mathbb{E}^{(x,y) \sim \mu}[\ell(f(x), y)] - \frac{1}{n} \sum_{i=1}^{n} \ell(f(x_i), y_i) \right| \leq O\left(\max\left(\sqrt{\frac{k}{n}}, L\sqrt{\frac{c\log(|\mathcal{F}|)}{nd}}, \sqrt{\frac{\log(1/\delta)}{n}}\right)\right).$$

*Proof.* Using McDiarmid's concentration inequality it is enough to bound the left hand side in expectation over $(x_i, y_i)$. Using the symmetrization trick, one reduces this task to upper bound

$$\mathbb{E}^{x_i, y_i, \sigma_i} \sup_{f \in \mathcal{F}} \frac{1}{n} \sum_{i=1}^{n} \sigma_i \ell(f(x_i), y_i).$$

Fixing the pairs $(x_i, y_i)$ and using the contraction lemma (see e.g., [SSBD14, Theorem 26.9]) the above quantity is upper bounded by $Rad_{n,\mu}(\mathcal{F})$ which concludes the proof. $\square$

Of course, one can again use an $\epsilon$-net to obtain an analogous result for continuously parametrized function classes. The law of robustness, now for a general loss function, follows as a corollary (the argument is similar to [Proposition 1, [BELM20]]). Let us point out that many papers have studied the Rademacher complexity of function classes such as neural networks (see e.g. [BFT17], or [YKB19] in the context of adversarial examples). The new feature of our result is that isoperimetry of the covariates yields improved generalization guarantees.

## Funding Acknowledgement

Funding in direct support of this work: NSF grant CCF-2006489, NSF graduate research fellowship, Stanford graduate fellowship. Additional revenues related to this work: internship employment by the second author at Microsoft Research.

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
