# A  Necessity of Polynomially Bounded Weights

In [BLN21] it was conjectured that the law of robustness should hold for the class of *all* two-layer neural networks. In this paper we prove that in fact it holds for arbitrary smoothly parametrized function classes, as long as the parameters are of size at most polynomial. In this section we demonstrate that this polynomial size restriction is necessary for bounded depth neural networks.

First we note that *some* restriction on the size of the parameters is certainly necessary in the most general case. Indeed one can build a single-parameter family, where the single real parameter is used to approximately encode all Lipschitz functions from a compact set in $\mathbb{R}^d$ to $[-1, 1]$, simply by brute-force enumeration. In particular no tradeoff between number of parameters and attainable Lipschitz constant would exist for this function class.

Showing a counter-example to the law of robustness with unbounded parameters and "reasonable" function classes is slightly harder. Here we build a three-layer neural network, with a single fixed nonlinearity $\sigma : \mathbb{R} \to \mathbb{R}$, but the latter is rather complicated and we do not know how to describe it explicitly (it is based on the Kolmogorov-Arnold theorem). It would be interesting to give similar constructions using other function classes such as ReLU networks.

**Theorem 4.** *For each $d \in \mathbb{Z}^+$ there is a continuous function $\sigma : \mathbb{R} \to \mathbb{R}$ and a sequence $(b_\ell)_{\ell \leq 2^{2^d}}$ such that the following holds. The function $\Phi_a$ defined by*

$$\Phi_a(x) = \sum_{\ell=1}^{2^{2^d}} \sigma(a - \ell) \sum_{i=1}^{2d} \sigma\left(b_\ell + \sum_{j=1}^{d} \sigma(x_j + b_\ell)\right), \qquad |a| \leq 2^{2^d} \tag{A.1}$$

*is always $O(d^{3/2})$-Lipschitz, and the parametrization $a \to \Phi_a$ is 1-Lipschitz. Moreover for $n \leq \frac{2^d}{100}$, given i.i.d. uniform points $x_1, \ldots, x_n \in \mathbb{S}^{d-1}$ and random labels $y_1, \ldots, y_n \in \{-1, 1\}$, with probability $1 - e^{-\Omega(d)}$ there exists $\ell \in [2^{2^d}]$ such that $\Phi_\ell(x_i) = y_i$ for at least $\frac{3n}{4}$ of the values $i \in [n]$.*

*Proof.* For each coordinate $i \in [d]$, define the slab $\mathtt{slab}_i = \{x \in \mathbb{S}^{d-1} : |x_i| \leq \frac{1}{100d^{3/2}}\}$ and set $\mathtt{slab} = \bigcup_{i \in [d]} \mathtt{slab}_i$. Then it is not difficult to see that $\mu(\mathtt{slab}) \leq \frac{1}{10}$. We partition $\mathbb{S}^{d-1} \backslash \mathtt{slab}$ into its $2^d$ connected components, which are characterized by their sign patterns in $\{-1, 1\}^d$; this defines a piece-wise constant function $\gamma : \mathbb{S}^{d-1} \backslash \mathtt{slab} \to \{-1, 1\}^d$. If we sample the points $x_1, \ldots, x_n$ sequentially, each point has probability at least $\frac{4}{5}$ to be in a new cell - this implies that with probability $1 - e^{-\Omega(n)}$, at least $\frac{3n}{4}$ are in a unique cell. It therefore suffices to give a construction that achieves $\Phi(x_i) = y_i$ for all $x_i \notin \mathtt{slab}$ such that $\gamma(x_i) \neq \gamma(x_j)$ for all $j \in [n] \backslash \{i\}$. We do this now.

For each of the $2^{2^d}$ functions $g_\ell : \{-1, 1\}^d \to \{-1, 1\}$, we now obtain the partial function $\tilde{h}_\ell = g_\ell \circ \gamma : \mathbb{S}^{d-1} \backslash \mathtt{slab} \to \{-1, 1\}$. By the Kirszbraun extension theorem, $\tilde{h}_\ell$ extends to an $O(d^{3/2})$-Lipschitz function $h_\ell : \mathbb{S}^{d-1} \to [-1, 1]$ on the whole sphere. The Kolmogorov-Arnold theorem guarantees the existence of an exact representation

$$\Phi_\ell(x) = \sum_{i=1}^{2d} \sigma_\ell\left(\sum_{j=1}^{d} \sigma_\ell(x_j)\right) \tag{A.2}$$

of $h_\ell$ by a two-layer neural network for some continuous function $\sigma_\ell : \mathbb{R} \to \mathbb{R}$ depending on $\ell$. It suffices to give a single neural network capable of computing all functions $(\Phi_\ell)_{\ell=1}^{2^{2^d}}$. We extend the definition of $\Phi_a$ to any $a \in \mathbb{R}$ via:

$$\Phi_a(x) = \sum_{\ell=1}^{2^{2^d}} \sigma(a - \ell)\Phi_\ell(x) \tag{A.3}$$

where $\sigma : \mathbb{R} \to \mathbb{R}$ satisfies $\sigma(x) = (1 - |x|)_+$ for $|x| \leq 2^{2^d}$. This ensures that (A.3) extends (A.2). To express $\Phi_a$ using only a single non-linearity, we prescribe further values for $\sigma$. Let

$$U = 2^{2^d} + d \cdot \max_{x \in [-1, 1], \ell \in [2^{2^d}]} |\sigma_\ell(x)|$$

so that $\left| \sum_{j=1}^d \sigma_\ell(x_j) \right| \leq U$ for all $x \in \mathbb{S}^{d-1}$. Define real numbers $b_\ell = 10\ell U + 2^{2^d}$ for $\ell \in [2^{2^d}]$ and for all $|x| \leq U$ set

$$\sigma(x + b_\ell) = \sigma_\ell(x).$$

Due to the separation of the values $b_\ell$ such a function $\sigma$ certainly exists. Then we have

$$\Phi_\ell(x) = \sum_{i=1}^{2d} \sigma \left( b_\ell + \sum_{j=1}^d \sigma(x_j + b_\ell) \right).$$

Therefore with this choice of non-linearity $\sigma$ and (data-independent) constants $b_\ell$, some function $\Phi_\ell$ fits at least $\frac{3n}{4}$ of the $n$ data points with high probability, and the functions $\Phi_a$ are parametrized in a 1-Lipschitz way by a single real number $a \leq 2^{2^d}$.

$\square$

**Remark A.1.** The representation (A.1) is a three-layer neural network because the $\sigma(a - \ell)$ terms are just matrix entries for the final layer.

**Remark A.2.** The construction above can be made more efficient, using only $O(n \cdot 2^n)$ uniformly random functions $g_\ell : \{-1, 1\}^d \rightarrow \{-1, 1\}$ instead of all $2^{2^\ell}$. Indeed by the coupon collector problem, this results in all functions from $\{\gamma(x_i) : i \in [n]\} \rightarrow \{-1, 1\}$ being expressable as the restriction of some $g_\ell$, with high probability.