# OpenReview forum: "A Universal Law of Robustness via Isoperimetry"
_NeurIPS.cc/2021/Conference — NeurIPS 2021 Oral_

### Official Review · Reviewer_Z64o · 2021-07-15

**Rating:** 10
**Confidence:** 5

**Summary:**

It is generally possible to interpolate train data as long as the number of parameters of our model exceeds the number of training points. The present paper asks how many parameters do we need if we want to interpolate in a smooth manner. If the smoothness is measured by Lipschitzness, the paper proves that one needs at least n d parameters to do so, where d is the ambient dimension (Theorem 1). The main assumptions are i) Lipschitz parametrization of the model with bounded parameters, ii) isoperimetric distribution of the train data, iii) positive noise level. This result, in particular, settles a conjecture by Bubeck, Li, and Nagaraj modulo assumption iii).

**Limitations And Societal Impact:**

yes

**Main Review:**

***Pros***

 - the proposed result is beautiful and truly interesting for the community. If indeed overparametrization (by a factor d) is necessary for smooth interpolation, this could have practical incidence on the type of model one tries to train on real datasets. This needs, of course, to be supported by other theoretical results and further experimental evidence. Note that such speculative discussion is extensively detailed in the paper (Section 1.1).
 - the theoretical framework is sound and simple enough so that one can imagine building upon it.
 - the exposition of the paper is very clear.

***Cons***

 - there seems to be a slight inaccuracy in the proof of Lemma 2.1. At the end of p. 5, when lower bounding \norm{Z}^2+2\langle Z,G-F\rangle + \norm{G-F}^2, I find \sigma^2 - \epsilon/6 + 2(-\epsilon/6)-2\langle Z,F\rangle + 0 (using l. 207 and \norm{G-F}^2\geq 0). Therefore the lower bound is \sigma^2-\epsilon/2-2\langle Z,F\rangle. The above display thus reads \langle F,Z\rangle \geq \epsilon /4, not \epsilon /3 and same thing in the statement of the lemma. I did not echo further, but it seems to change the way \epsilon has to be cut before using Lemma 2.1 in the subsequent derivations. To be clear: this is not a major mistake, the proof certainly still holds, but the numerical constants could change. I will increase my score if the rebuttal makes the changes to be made precise (or if I missed something here).

***Minor comments and typos***

 - l. 149: missing 2 in the definition of sub-Gaussian. Also t is not defined in this definition.
 - l. 206: comma inside the parenthesis in the definition of G

**Time Spent Reviewing:**

10

---

> ### Author Response · Authors · 2021-08-05
> **Response to Reviewer Z64o**
>
> We thank you for your kind words. We find it particularly encouraging that you agree with us on the potential practical significance of our result.
>
> You are also correct that epsilon/3 should be epsilon/4 in Lemma 2.1., thank you for noticing this inaccuracy! The downstream effect of this change is in (2.3) and (2.4) (and their rewriting for k>1) the terms epsilon/6 should be epsilon/8. Thankfully no further change is needed, as the upper bound on those terms already had bigger constants than necessary (the terms 8^3 and 8^4 in the exponentials).

---

> > ### Comment · Reviewer_Z64o · 2021-08-26
> > **After rebuttal**
> >
> > Thank you for the answer, it is a good thing that there was a small margin in the constants further down.

---

> > > ### Author Response · Authors · 2021-08-26
> > > **Thank you!**
> > >
> > > Thank you! Gentle reminder that your review mentioned: "I will increase my score if the rebuttal makes the changes to be made precise ".
> > >
> > > Best,
> > > The Authors

---

### Official Review · Reviewer_Xj7d · 2021-07-16

**Rating:** 8
**Confidence:** 3

**Summary:**

This paper shows that if one wants to interpolate the data smoothly, the number of parameters required is $d$ times more than mere interpolation. More precisely, they show that for any function class smoothly parametrized by $p$ parameters, and for any $d$-dimensional dataset satisfying mild regularity conditions, any function in this class that fits the data below the noise level must have its (Euclidean) Lipschitz constant larger than $\sqrt{nd/p}$.

To demonstrate this, the authors show that for random labels, if the distribution satisfies isoperimetry then either the 0 level or 1 level set has probability upper bounded by $\exp(-d/C^2)$ where $C$ is the Lipschitz constant of $f$. A union bound then gives an upper bound on the probability of a function fitting the data.


**Limitations And Societal Impact:**

This is a theoretical paper and of limited social impact.

**Main Review:**

I think this is a good paper and vote to accept it. I find the result interesting and the paper proves a prior conjecture in the case of polynomially bounded weights. I think the proof is intuitive and clearly presented.

**Time Spent Reviewing:**

2

---

> ### Author Response · Authors · 2021-08-05
> **Response to Reviewer Xj7d**
>
> We thank you for your vote to accept the paper. We also would like to take this opportunity to emphasize that this work goes beyond "proving the prior conjecture for polynomially bounded weights". Indeed, the universal law of robustness is the first result to *provably* show that overparametrization is *necessary* in a very broad context. Most importantly, the prior conjecture was much more narrow and did not talk about multilayer neural networks, while this work applies to this important class of functions.

---

### Official Review · Reviewer_gUHM · 2021-07-16

**Rating:** 6
**Confidence:** 4

**Summary:**

The paper describes a universal law of robustness via isopemitry, which states that smooth interpolation requires d times more parameters than usual interpolation. The authors then include a discussion of their results for the MNIST and ImageNet datasets, and go through the proof of the main results. The authors conclude with a discussion of their results for deep learning.

**Limitations And Societal Impact:**

No discussions on societal impact is available.

**Main Review:**

1. Originality

The tasks and/or methods are not new. The work is a direct follow-up of [BLN21] and the proof techniques regarding isometry and concentration are not new. Nevertheless, the main result is new and fairly interesting.

2. Quality

The submission is technically sound and the main theorems make sense. A part that may need more explanation or refinement is the speculative discussion with MNIST and ImageNet. The main results are stated with constants, and even though the law of robustness "predicts any such model must have at least nd parameters", it is not clear if it is 100nd or 0.1nd; perhaps some additional clarification on the constant would be helpful. Another part that I feel can be beneficial to add in the paper is a discussion on cases where the main results are tight; i.e., conditions where an if and only if result holds. Additional discussion on the weaknesses of the work would also be helpful.

3. Clarity

The paper is nicely written and organized. I would think it would be more beneficial to use more of the pages to discuss the implications/significance of the results and put the proof in the supplementary (since the proof takes up almost half of the main text). Some additional comments regarding typos are included in the Section 5 of this main review. Two minor comments on clarity:

a. In Theorem 1, where perhaps it would be nice to add the dependence of epsilon in the equation.

b. In the abstract, particularly the last two sentences, "We prove this universal law of robustness... this law was conjectured in prior work by Bubeck, Li and Nagaraj" gave me the impression that the work confirmed this conjecture for 2-layer NNs and Gaussian covariates, however, in Section 1.2 the authors clarify that "Our results does not actually prove their conjecture,...". I think it may be a good idea to change the text in the abstract to prevent this kind of subtle misinterpretation.

4. Significance

The results continues work on conjectures from prior work due to Bubeck, Li and Nagaraj, and from this theoretical perspective the results are important. However, due to the pure theoretical nature of this work and plenty of the main text occupied by the proof, practical significance is a bit lacking, and perhaps this may be addressed in further revisions.

5. Typos

A minor typo is in Line 36, "theoreof" perhaps should be "thereof".

Update after reviewing comments: Increasing score to 6.

**Time Spent Reviewing:**

3 hours

---

> ### Author Response · Authors · 2021-08-06
> **Response to Reviewer gUHM**
>
> We thank you for your careful reading of the paper. Below we give answers to your questions/comments.
>
> 2. Quality
>
> -- You make a good point about the numerical constants and we will clarify this in the final version. Our main goal in the formal parts of the paper (e.g., Theorems 2/3 and their proofs) was to give the most insightful calculations to obtain the correct *scaling law* (namely, p ~ nd/Lip^2). This comes at the expense of somewhat large numerical constants (e.g., 2^9 in Theorem 3). However we believe that with more effort one could obtain numerical constants actually close to 1. The evidence for this is that in the toy case described at the end of page 4 we actually have constants close to 1. Moreover we note that, in the speculative section, all calculations are done by only keeping the "order of magnitude" of the various terms that appear. The "true" numbers might indeed be somewhere between 0.1 and 10 times of what we predict.
>
> -- You make another excellent point regarding tightness, we will add more details regarding this in the final version. It turns out that the law of robustness *is* tight in the sense that there exists a function class satisfying all our assumptions and such that one can get perfect fit and Lipschitz constant bounded by sqrt(n d / p). Please see our response to Reviewer pXqe for an outline of the proof of this result (paragraph starting with "We apologize"; see also there the paragraph that starts with "Regarding specific function classes" which is relevant to your question too).
>
> 3. Clarity
>
> We did our best to convey the implications/significance of our result in the introduction. If the reviewer has any suggestions on what to add we would love to hear it. We also would like to take this opportunity to clarify that, from our perspective, the proof is an integral part of the significance of the paper. Indeed, while for now almost a decade overparametrization has been a central part of the deep learning framework, there has been to date no general *mathematical proof* that such overparametrization is necessary (in a broad/non-artificial setting). A key strength of our work is that we found such a broad setting where we can mathematically prove the necessity of overparametrization.
>
> a. Great suggestion, we will add the dependence on epsilon.
> b. This is a delicate point, and we have tried to use careful wording whenever we refer to the work of Bubeck, Li and Nagaraj. In essence, the conjecture of BLN remains open because in our work we need to assume polynomially bounded weights. However we also prove that this polynomial weight assumption is *necessary* as soon as one considers neural networks with >1 hidden layers. Thus, in a sense, while the conjecture of BLN remains open, at this point it is arguably a pure math problem, without obvious implications for machine learning. In other words, our work resolves the BLN conjecture for the case of *practical interest* (and in fact we do much more, see next point).
>
> 4. Significance
>
> An important point that we would like to clarify is that our work does much more than prove (a special case of) the BLN conjecture. Indeed, in the latter work the authors only considered the case of a two layer neural network, while our work applies to arbitrary function classes (including e.g., resnets, transformers, etc....). We also believe that this might be one the rare cases where *pure theoretical work* actually has practical significance, in the sense that we make concrete predictions about the scaling necessity for robustness (see for example the review by Reviewer Z64o who concurs with this point).

---

> > ### Comment · Reviewer_gUHM · 2021-08-25
> > **Update after reading comments**
> >
> > Thanks for the detailed responses addressing my concerns and questions---I am increasing my score to 6. The comments on significance in the reply as well as replies to other reviewer comments are quite nice and I believe it would be helpful to add them into further versions of the paper.

---

### Official Review · Reviewer_pXqe · 2021-07-16

**Rating:** 7
**Confidence:** 4

**Summary:**

This paper provides a lower bound on the *global* Lipschitz constant (with respect to the 2-norm) of any function within a parameterized function class that fits data below the noise level that scales inversely in the number of parameters p, showing that overparameterization is *necessary* to ensure a smooth fit of noisy training data. These lower bounds hold under the following assumptions/caveats:

- the d-dimensional input data satisfies the assumption of c-isoperimetry, which includes the Gaussian measure, strongly log-concave measures and distributions over d-dimensional manifolds with a positive Ricci curvature.

-- the parameters ("weights") of the function class are polynomially bounded in (n,d).

-- the training data has positive label noise, and the function is constrained to fit below the noise level (which includes the case of interpolation of training data).

-- the smoothness (Lipschitz) constant is evaluated with respect to the 2-norm.

The paper contextualizes these results in the context of deep neural networks and provides two lower bounds on the Lipschitz constant that (in addition to the scaling given in n, d, and p) scale inversely either in the depth or the product of spectral norms of layers. The latter is a well-known upper bound on the Rademacher complexity of deep neural networks. Both results show an additional dependence on the number of parameters p that is required to ensure small Lipschitz constant --- such a dependence is not present in typical Rademacher complexity bounds used to upper bound clean generalization error.

**Limitations And Societal Impact:**

Yes, the authors have adequately addressed the limitations and potential negative societal impact of their work. One suggestion: the authors are encouraged to provide a contextualization of their results with the following work, e.g. in the broader discussion in Section 1.1:

"Rademacher complexity for adversarially robust generalization", by Dong Yin, Kannan Ramchandran and Peter L Bartlett.

This paper takes a different approach and upper bounds the Rademacher complexity of adversarially robust generalization error for specific functions (and also uncovers extra dimension dependencies).



**Main Review:**

Strengths of submission:

-- Establishing fundamental limits on adversarial robustness of function classes is an important open problem. This paper makes progress towards this mathematically challenging goal (albeit by lower bounding the global Lipschitz constant, whose precise relationship to the adversarial error remains unclear).

-- The proof technique is relatively elementary, but innovative. In particular it makes elegant use of the isoperimetry property (which by definition is connected to Lipschitz-ness). Because of the ultimate application of parameter counting, it is also very generally applicable to arbitrary parametric function classes and shows insightfully that the number of parameters fundamentally bottlenecks a smoothness guarantee subject to fitting a non-zero amount of noise.

-- This proof technique could be useful to study the generalization error of interpolating solutions for generic function classes beyond the adversarial examples problem.

Weaknesses of submission:

-- As discussed in Section 1.1 of the paper, it is very unclear whether the global Lipschitz constant is fundamentally related to the adversarial generalization error: intuitively, their relationship would depend on more fine-grained quantities such as the *distribution* of the Lipschitz constant over the data domain. In fact one can construct toy scenarios using 1-D data and polynomial features for which the global Lipschitz constant could worsen with overparameterization, but generalization error (clean or adversarial) might improve depending on the overall distribution of the local Lipschitz constants. Question for authors: do you think there is any possibility of studying these more fine-grained properties in the future, that might be more reflective of adversarial error, as a consequence of this approach?

-- The results would also be stronger if matching, or at least close upper bounds could be provided, i.e. *there exists* a function within the function class that has sufficiently small Lipschitz constant that decays with overparameterization. This would be required to make the definitive case that overparameterization actually allows us to find functions with a small global Lipschitz constant (which is in fact sufficient for low adversarial error). Alternatively, this would shed light on whether a low Lipschitz constant is reasonable to expect and *necessary* for adversarial robustness; either way, it would be of interest. Perhaps this is possible via an anti-concentration approach, but it would also require characterizing the size of the set of globally L-smooth functions in the function class (which is not needed for the lower bound as one can simply upper bound this by the size of the function class itself). Do the authors think their techniques could be used for an upper bound in the future for the same problem; if so, how?

-- The techniques are currently presented for Lipschitz constant with respect to l2-norm: the more standard quantity in the adversarial robustness literature is the infinity-norm. The authors mention in Section 1.1 that they believe that a variant of the law of robustness would also hold for the infinity-norm, but they do not provide details on this (in particular I am not sure if the techniques will be similar for the case of distributions over manifolds).

Minor caveats/additional context that could be added in the paper:
-- Because of the centrality of the union bounding approach to the proof, the techniques appear unlikely to shed light on the smoothness of specific functions that interpolate training data and arise in practice (e.g. variously regularized neural nets such as minimum-norm, minimum-path-norm, etc).

-- The results critically constrain the function class to be fitting a non-zero amount of noise, which may not always be the case in practice. A short discussion on whether these results might imply a small Lipschitz constant on regularized function classes and/or noiseless data might be a good addition to the paper.

---- after response ----
thanks to the authors for their detailed and well-thought-out response. I've increased my score accordingly. Please see comments below for details.

**Time Spent Reviewing:**

3

---

> ### Author Response · Authors · 2021-08-05
> **Response to Reviewer pXqe**
>
> We thank you for your very detailed review. Your comments get to the heart of the matter, and in fact several of your questions raise challenging open problems that we have thought about, and which we believe might constitute interesting research directions with potentially great practical significance. Below we give detailed responses to several of your comments.
>
> Response to weaknesses:
>
> -- Studying more fine-grained properties is one such example of a fantastic open problem, with potentially great significance in practice. We note however that the situation is quite subtle mathematically: natural alternative measures of smoothness such as Sobolov norms (see last paragraph in Section 1.2) or the Barron norm for the case of a two layer net (see [Bubeck, Eldan, Lee, Mikulincer, NeurIPS 2020]) do NOT admit a tradeoff a la law of robustness. That is, for those measures of smoothness it is possible to build networks with as few parameters as information theoretically possible (i.e., p = O(n)) while also being as smooth as possible (either in the sense of a Sobolev norm or the Barron norm). From our point of view, one of the strengths of the law of robustness is that we found a setting where the tradeoff exists under broad conditions. We believe other "laws" like this exist, but they might be difficult to come by.
>
> -- We apologize for not making this clear in the submission, but there is actually a rather simple argument to show a matching upper bound to the law of robustness (we will add it in the final version) by using a function class consisting of sums of “bump functions” https://en.wikipedia.org/wiki/Bump_function. First observe that with n*(d+1) parameters one can get perfect fit with O(1) Lipschitz constant by superposing bump functions around each data point (each such bump can be defined with d+1 parameters, d for the center and 1 for the height). The fact that this is O(1)-Lipschitz is directly related to the fact that the distance between any two data points is Omega(1) (here one needs n=poly(d), which is needed for mere existence of a O(1)-Lipschitz map, see Remark 1.1.). Now if one has only p parameters, with potentially p<<n*d, one can first randomly project into D=p/n dimensions, so that the points are now Omega(sqrt(D/d)) close, and repeat the above argument there. The number of parameters used is n*(D+1) = p+n, and the Lipschitz constant is O(sqrt(d/D)) = O(sqrt(n*d / p)), thus matching the law of robustness lower bound.
>
> -- Regarding other norms: the essence of our proof is actually not tied to l2, in the sense that everything goes through as long as one assumes isoperimetry in the norm of interest. The difficulty however becomes to find interesting/realistic cases in which isoperimetry provably holds for these potentially more complicated norms.  In some cases this could be a wonderful mathematical open problem (e.g., for Wasserstein norm?).
>
> -- Regarding specific function classes that arise in practice: since our model of parametrized function classes is quite general, it includes practical function classes (e.g., resnet, transformers, etc...). On the other hand, it is completely plausible that *stronger* laws could be proved when some regularization is added. We only know of one such example, published in Theorem 7 of [Bubeck, Li, Nagaraj, COLT 2021], where it is shown that for two layer nets with a quadratic activation function, one gets an additional multiplicative term in the lower bound of sqrt(d / rank), which shows that regularizing for "rank" actually induces a stronger law of robustness. Establishing the full scope of these "stronger laws" is yet another fantastic open problem.
>
> -- Regarding noisy/noiseless data: this is a conceptually very interesting point. We briefly touch on it in the fourth paragraph of Section 1.1. We believe there is room to prove a law of robustness for a noiseless setting, but it will require to add a few more ingredients, in particular *algorithms* might enter the picture here.
>
> -- Thank you for the reference suggestion, we will add a comment/citation to this work of Yin, Ramchandran and Bartlett.
>
>
> Minor correction to your summary: you mention that we prove two lower bounds which "scale inversely either in the depth or the product of spectral norms of layers". We wanted to bring your attention to the fact that for the case of the product of spectral norms, we only depend on the *logarithm* of this product. Thus this is a much stronger lower bound than the one you would get from prior work (e.g., Foster, Telgarsky, Bartlett) showing an upper bound on the Rademacher complexity in terms of the product of spectral norms.
>
>
> Minor addition to your third point on "strength of the submission": we wanted to bring your attention to the fact that we already worked out one such generalization consequence in the supplementary material (Appendix B). Namely our argument can be interpreted as showing that function classes with p parameters and L Lipschitz functions have a generalization error upper bounded by L sqrt( p / (d * n) ) when the covariate distribution is a mixture of isoperimetric measures. In essence, this allows you to argue that n data points effectively act as n*d data points from a generalization perspective.

---

> > ### Comment · Reviewer_pXqe · 2021-09-01
> > **Thank you for your detailed response!**
> >
> > I would like to sincerely thank the authors for their detailed and well-thought-out response. I would recommend adding as much of this discussion as is possible in the camera-ready version if the paper is accepted.
> >
> > I am increasing my score as I think it's apparent that the authors have thought through their assumptions and framework, and I think the result is interesting and non-trivial.
> >
> > Please consider adding the matching upper bound as a formal result (with proof): this will likely be of interest to readers.

---

### Decision · Program_Chairs · 2021-09-27

**Decision:**

Accept (Oral)

**Comment:**

This paper adds an interesting ingredient to the substantial literature on interpolating / memorizing data using neural networks by asking how things change if the neural network is required to be smooth (Lipschitz). The results essentially show that there is a cost to smoothness, appearing as a multiplicative factor of d in the number of parameters of the model. Smoothness of the model loosely corresponds to robustness against adversarial examples, and thus, the results shed light on the empirically observed phenomenon that sufficiently large models can be made more robust. Significantly, they give fairly precise numerical predictions that may help to guide the practice of deep learning.